# The Lung Microbiome: A New Frontier for Lung and Brain Disease

**DOI:** 10.3390/ijms24032170

**Published:** 2023-01-21

**Authors:** Jiawen Chen, Ting Li, Chun Ye, Jiasheng Zhong, Jian-Dong Huang, Yiquan Ke, Haitao Sun

**Affiliations:** 1Clinical Biobank Center, Microbiome Medicine Center, Department of Laboratory Medicine, Zhujiang Hospital, Southern Medical University, Guangzhou 510280, China; 2Neurosurgery Center, The National Key Clinical Specialty, The Engineering Technology Research Center of Education Ministry of China on Diagnosis and Treatment of Cerebrovascular Disease, Guangdong Provincial Key Laboratory on Brain Function Repair and Regeneration, The Neurosurgery Institute of Guangdong Province, Zhujiang Hospital, Southern Medical University, Guangzhou 510282, China; 3School of Biomedical Sciences, Li Ka Shing Faculty of Medicine, University of Hong Kong, Hong Kong, China; 4Chinese Academy of Sciences (CAS) Key Laboratory of Quantitative Engineering Biology, Shenzhen Institute of Synthetic Biology, Shenzhen Institutes of Advanced Technology, Chinese Academy of Sciences, Shenzhen 518055, China; 5Clinical Oncology Center, Shenzhen Key Laboratory for Cancer Metastasis and Personalized Therapy, The University of Hong Kong-Shenzhen Hospital, Shenzhen 518055, China; 6Guangdong-Hong Kong Joint Laboratory for RNA Medicine, Sun Yat-Sen University, Guangzhou 510120, China

**Keywords:** lung microbiome, chronic lung diseases, lung cancer, multiple sclerosis, lung–brain axis

## Abstract

Due to the limitations of culture techniques, the lung in a healthy state is traditionally considered to be a sterile organ. With the development of non-culture-dependent techniques, the presence of low-biomass microbiomes in the lungs has been identified. The species of the lung microbiome are similar to those of the oral microbiome, suggesting that the microbiome is derived passively within the lungs from the oral cavity via micro-aspiration. Elimination, immigration, and relative growth within its communities all contribute to the composition of the lung microbiome. The lung microbiome is reportedly altered in many lung diseases that have not traditionally been considered infectious or microbial, and potential pathways of microbe–host crosstalk are emerging. Recent studies have shown that the lung microbiome also plays an important role in brain autoimmunity. There is a close relationship between the lungs and the brain, which can be called the lung–brain axis. However, the problem now is that it is not well understood how the lung microbiota plays a role in the disease—specifically, whether there is a causal connection between disease and the lung microbiome. The lung microbiome includes bacteria, archaea, fungi, protozoa, and viruses. However, fungi and viruses have not been fully studied compared to bacteria in the lungs. In this review, we mainly discuss the role of the lung microbiome in chronic lung diseases and, in particular, we summarize the recent progress of the lung microbiome in multiple sclerosis, as well as the lung–brain axis.

## 1. Introduction

Trillions of microbes have evolved alongside humans and now live on and within them, influencing human health and disease [1,2]. Owing to the restriction of traditional culture-dependent methods, it has been thought that the lung is a sterile organ in its healthy state, and that there are microbiomes only in the disease state [3,4,5]. This belief seriously hinders the study of the lung microbiome [3]. For more than a decade, the bacterial community in the lungs has increasingly attracted the spotlight with the development of culture-independent techniques. Using culture-independent techniques such as 16S RNA gene sequencing and shotgun metagenomic sequencing, it has been found that microbiomes also exist in healthy lungs, breaking the long-held belief that the lung is a sterile organ [4,6]. This is consistent with the natural physiology of the lungs, as each breath is exposed to the environment and the lungs are closely connected to the upper airway [3]. The microbial biomass of the lungs is low compared to microbiota in other sites, but the ecology of the lungs’ microbiome is diverse and complex [7,8]. When in a diseased state (such as chronic lung disease or lung cancer), there is a disruption of microbial homeostasis, with compositional or functional changes in the microbial community [9,10]; that is, the relative abundance of microbial components within the community changes. Emerging studies have shown that the lung microbiome may play a potential role in the pathogenesis of chronic lung diseases and lung cancer [4,11,12]. In addition to lung disease, recent studies have shown that the lung microbiome can also affect brain autoimmunity [13,14]. This article mainly discusses the role of the lung microbiome (mainly bacteria) in chronic lung diseases and, in particular, summarizes the current research findings on the lung microbiome in multiple sclerosis.

## 2. Methods of Analyzing the Lung Microbiome

Culture-based techniques have primarily focused on single isolated species, whereas sequence-based approaches can capture entire communities. In recent years, molecular identification techniques have also been widely used to explore the lung microbiome, with 16S rRNA gene sequencing being the most widely used method for assessments of community composition. The 16S gene encoding bacterial ribosomal RNA is a small and highly conserved locus in bacterial DNA, containing nine hypervariable regions [15]. Although full-length 16SrRNA gene sequencing provides greater taxonomic definition [16], detection of the lung microbiome is more commonly performed for one or more of the nine hypervariable regions (V1–V9) [17,18]. In addition, the definition of the microbiome includes not only bacteria but also fungi and viruses. The detection of fungi is also based on the sequencing of targeted regions in the ribosomal locus, such as the 18S rRNA gene or internal transcribed spacer (ITS) [19]. However, because viruses lack conserved nucleic acid sequences, shotgun metagenomics—which sequences the entire nucleic acid extracted from the sample—is still primarily used in the investigation of viruses in lung diseases [20,21].

The 16S rRNA gene sequencing approach is mainly used to describe the lung microbiome’s taxonomy, enabling analysis of microbial community features such as diversity and relative abundance. However, it is limited by the fact that most bacteria will only be identified by genus or family—rarely at the species level—and lack functional information. Shotgun metagenomics gene sequencing can capture functional information about a microbial community (bacterial, fungal, or viral), allowing researchers to investigate antimicrobial resistance or virulence genes [22]. Notably, neither shotgun metagenomics nor 16S rRNA gene sequencing can distinguish between live and dead bacteria. Traditional culture-based methods for assessing bacterial viability have proven to be cost-effective. However, many bacterial species have been found to exist in a viable but non-culturable state [23]. Metatranscriptomics is more suitable here, as it studies the gene expression of microbes within natural environments [24]. In addition, the application of mass spectrometry to study microbe–host protein interactions is also a current trend [25]. Hence, to conduct a more comprehensive analysis of the composition of the lung microbial community and its interactions with the host, a combination of culture-based and culture-independent methods should be applied.

## 3. The Healthy Lung Microbiome

Numerous studies have been conducted on the effects of gut bacteria on human disease and health, and in recent years the lung microbiome has attracted increased interest. Traditionally, it was believed that a healthy lung—including the distal alveoli—was a sterile organ [3]. In reality, the lungs are constantly exposed to microbes that are floating in the air and in the upper airways. About 80% of the microbes inhabiting the human body cannot be found via conventional culture-based techniques, posing a great challenge to proving the existence of the microbiome in the lungs [4]. The abundant microbiome in the lungs was reported for the first time in 2010 as culture-independent techniques evolved. Subsequently, many studies have also confirmed the existence of a lung microbiome [7,26]. It is now thought that the lungs are not a sterile environment but, rather, that they contain a rich microbiome that plays a significant role in health and disease. In the lung microbiome, the dominant phyla are *Firmicutes* and *Bacteroidetes*, and the main bacterial genera are *Prevotella*, *Porobacteria*, and *Streptococcus* [7,27]. The healthy lung microbiome comprises a complex and diverse bacterial community, with a low biomass of only 10^3^ to 10^5^ bacteria per gram of tissue [4,7,28]. Recently, the understanding of the lung microbiome has become increasingly clear, but it lags far behind that of the gut microbiome. Since healthy lung tissue samples are usually difficult to obtain, there are three main types of samples used to analyze the lung microbiome: endotracheal aspirates (ETAs), bronchoalveolar lavage (BAL), and sputum [3,6]. However, these methods may contain contamination, presumably from the upper respiratory tract; therefore, a rigorous control design is required to ensure the authenticity of the results [3,29]. In Figure 1, we summarize the process of the detection of lung microorganisms.

Mounting evidence suggests that the composition of the lung microbial community is mainly determined by three factors: migration, elimination, and relative reproduction rates, which are controlled by local environmental and growth conditions [3,4,7,10,28]. A number of studies found that the composition of the oral microbial community was nearly the same as that of the lung microbial community. The oral microbial community had a high biomass, while the lung microbial community had a low biomass [3,7,10,30,31]. Therefore, it is thought that the lung microbiome is mainly migrated into the lungs from the oral microbiota via micro-aspiration and mucosal translocation, which occur in healthy individuals [4,7,32]. The clearance of the lung microbiome is a dynamic process that mainly includes mucociliary clearance and immune defense systems. Healthy airways contain ciliated epithelia that propel microbes proximally. In addition, coughing can also expel microbes from the respiratory tract [3,28]. The respiratory tract and lungs have both innate and adaptive immune responses that can recognize and eliminate microbes. As described by studies, the lung is a highly heterogeneous environment that can affect the growth and reproduction of microorganisms and, in turn, impact the composition of the microbiome. Heterogeneity includes oxygen tension, pH, relative blood perfusion, relative alveolar ventilation, temperature, epithelial cell structure, deposition of inhaled particles, and the concentration and activity of inflammatory cells [5,7,28]. However, studies have shown less variability in the lung microbiome, suggesting that the composition of the lung microbiome is mainly determined by migration and clearance rather than relative reproduction rates [7]. In addition, whether there are geographic variations in the lung microbiome remains unclear and requires further study.

## 4. The Lung Microbiome in Lung Disease

In recent years, as knowledge of lung microbes has grown, their role in the development of lung diseases—such as chronic obstructive pulmonary disease (COPD), asthma, and lung cancer—has been gradually recognized. Many studies have shown that the lung microbiome can result in disease or increase disease susceptibility. This section focuses on the relationship between the lung microbiome and lung disease.

### 4.1. Chronic Obstructive Pulmonary Disease (COPD)

Chronic obstructive pulmonary disease (COPD) is one of the most common respiratory diseases, characterized by persistent symptoms and impaired lung function caused by airway inflammation, small airway occlusion, and emphysematous destruction of parenchyma [11,33,34,35,36]. While being primarily associated with smoking, it is also connected with other factors, including indoor smoke exposure, pollution, early respiratory infections, and genes [33,37]. The exacerbation of chronic obstructive pulmonary disease (COPD) has significant impacts on human health status and can even present a major threat to life [34,35]. Over the years, bacterial infection has been thought to be a major trigger for the exacerbation of COPD [33,38,39]. Bacterial colonization of the lower airway is fairly common in patients with stable COPD [40,41]. Through 16sRNA sequencing, a unique bacterial community was identified in the airways of patients with COPD. In the airways of patients with stable COPD, the predominant bacteria were *Moraxella catarrhalis*, *Streptococcus pneumoniae*, and *Pseudomonas aeruginosa* [33,40,42]. One longitudinal study, which collected samples from patients with COPD in different states, found that compared with stable COPD there were no significant differences in the composition of the lung microbiome according to 16sRNA gene sequencing in patients with COPD during acute exacerbations [33]. However, there was an overall reduction in the bacterial alpha diversity of the lung microbiome (i.e., microbial diversity within the sample) and an increase in bacterial load [41]. More specifically, acute exacerbations of COPD were associated with a significant increase in *Proteobacteria* and a significant reduction in *Bacteroidetes* and *Firmicutes* [40,41,43]. However, a prospective longitudinal cohort study of patients with COPD found no increase in bacterial load in patients with acute exacerbations of COPD. It has been hypothesized that the cause of acute exacerbations in patients with COPD is pre-existing bacteria and not an increase in bacterial load. The mechanism by which a pre-existing strain causes deterioration may be by altering its antigenic structure to evade the immune response, or by altering the airway environment through another infection [44]. The results from these studies are not entirely consistent.

Recently, Yan et al. [45] presented an in-depth profiling of the sputum metagenome, metabolome, host transcriptome, and proteome from 99 patients with COPD and 36 healthy individuals in China to show a landscape of airway microbe–host interactions in COPD. It was found that the relative abundances of *Moraxella catarrhalis* and *Pseudomonas aeruginosa* were most increased in COPD, while *Prevotella* intermedia was most depleted in COPD compared to healthy individuals. The increased abundance of these bacteria may increase systemic inflammation to cause COPD exacerbations [33]. One study indicated that, in patients with acute COPD exacerbations with *P. aeruginosa* infection, the level of IL-17 is increased [46]. IL-17 plays a detrimental role in the pathogenesis of *P. aeruginosa* airway infection during acute exacerbations of COPD [46]. In addition, integrated multi-omics for COPD microbe–host interactions indicated that *Lactobacillus* produces indole-3-acetic acid (IAA) that modulates lung function decline, inflammation, and apoptosis [45]. It has also been suggested that IAA inhibits apoptosis through IL-22-mediated macrophage–epithelial crosstalk [45]. The study indicated that derivatives of lung microbes may be involved in the development of COPD. However, more research is needed to demonstrate the specific role of the lung microbiome in COPD.

### 4.2. Asthma

Asthma is a common chronic respiratory disease that contributes to a significant global disease burden. Risk factors for asthma include genetics, early-life respiratory disorders (especially respiratory syncytial virus), airborne environmental exposure, and atopic sensitization [47,48,49,50,51]. One of the characteristics of asthma is chronic airway inflammation. Evidence has indicated that microbial airway colonization may play a role in the process of chronic inflammation [50]. A number of studies have confirmed the presence of bacteria in the lungs of patients with asthma, and the five major bacterial phyla include *Firmicutes*, *Proteobacteria*, *Actinobacteria*, *Fusobacteria*, and *Bacteroidetes*, of which the first three phyla account for more than 90% of the total sequences [52]. In addition, compared with controls, the bacterial burden of the lung microbiome was elevated in patients with asthma, with a marked increase in the numbers of *Proteobacteria*, containing the important potential pathogens *Haemophilus*, *Moraxella*, and *Neisseria* [53]. In addition to bacterial load, there is an increase in bacterial diversity in asthma patients. Although these pathogens are increased in patients with asthma, these associations do not yet establish causality between the presence of these pathogens and asthma [53]. Denner et al. [54] collected bronchial brush fluid and bronchoalveolar lavage fluid from 39 asthma patients and 19 control subjects and found that the main features of chronic asthma (i.e., airway responsiveness and airflow obstruction) were associated with changes in microbial burden, diversity, and composition. Similar to COPD, asthma is a heterogeneous chronic disease that can be divided into eosinophilic and non-eosinophilic inflammatory phenotypes correlated with the composition of the lung microbiome [54,55]. Studies have shown that both the diversity and uniformity of bacteria are lower in asthmatic patients with the neutrophil phenotype than in those with the eosinophil phenotype. In addition, the study also identified bacterial taxa that differed in the composition of the microbiome in the neutrophil inflammatory phenotype, where the abundance of *Haemophilus* and *Moraxella* taxa was higher. In contrast, *T. whipplei* abundance increased in the eosinophil granulocyte inflammatory phenotype [50,55]. There was a significant difference in airway bacterial burden between type 2 high and type 2 low patients with asthma, and it was found that several lung cytokines’ levels were significantly increased and sputum bacterial diversity was reduced in patients with type 2 low asthma [56]. In summary, differences in the airway microbiome are strongly associated with host inflammatory responses in asthma. Having a better understanding of the connections between different asthma phenotypes and the lung microbiome will help researchers to create more specific therapeutic strategies.

### 4.3. Lung Cancer

Lung cancer is the leading cause of cancer-related death throughout the world. The main pathological subtype of lung cancer is non-small-cell lung cancer, which can be divided into three main subtypes: adenocarcinoma, squamous-cell carcinoma (SCC), and large-cell carcinoma [57,58,59]. Lung cancer can be promoted by multiple factors, including chemical carcinogens, chronic inflammation, and bacterial and viral infections [3,58,60]. Although the impact of the gut microbiome on lung cancer has been extensively explored, few studies have investigated the interaction between the lung microbiome and lung cancer. Until recently, many studies have documented an association between the lung microbiome and lung cancer. Studies have shown that non-smoking lung cancer patients with chronic airway inflammation—such as chronic obstructive pulmonary disease (COPD), asthma, pneumonia, or tuberculosis—are at a higher risk of lung cancer, which suggests that the lung microbiome may play a role in the development and progression of cancer [3,58]. Jin C et al. [61] found that a germ-free genetically engineered murine model of human lung adenocarcinoma exhibited substantially delayed tumor growth compared with age-matched specific-pathogen-free (SPF) controls. They therefore proposed that commensal microbes promote the development of lung cancer. Another study, which collected protected bronchial brush samples from 24 lung cancer patients with unilateral lobar masses and 18 healthy patients, found that the alpha diversity of the lung microbiome in patients with lung cancer significantly decreased and gradually declined from healthy to non-cancerous to cancerous sites, indicating a significant change in the lung microbiome during the development of lung cancer [62]. The research by Peters et al. [63] reached the same conclusion; however, in this study, the microbial diversity and composition of lung cancer tissue were found not to be associated with recurrence-free survival (RFS), suggesting that the lung microbiome does not influence the recurrence of lung cancer. The composition of the lung microbiome is also closely related to the stage of lung cancer. Studies have shown that compared with patients with stage I–IIIA disease, patients with stage IIIB–IV NSCLC are more likely to show enrichment of the lower airway microbiota with oral commensals such as *Streptococcus*, *Prevotella*, and *Veillonella*, suggesting that the lung microbiome is associated with clinical prognosis [64]. In addition to tumor development and clinical prognosis, the microbiome—especially the intratumoral microbiome—may also have a close relationship with tumor metastasis. As recently noted, the tumor-resident intracellular microbiota promotes metastatic colonization in breast cancer. It was found that certain intracellular bacteria enhance the survival of circulating tumor cells through cytoskeleton reorganization in a spontaneous murine breast tumor model, MMTV-PyMT, in which the primary tumor was frequently accompanied by lung metastases [65]. Previously, *Fusobacterium* was found to be present in colorectal cancer tumor tissue and corresponding metastatic liver cancer tissue by culture, quantitative polymerase chain reaction (qPCR) analysis, and RNA ISH analysis [66]. The team also observed the consistency of *Fusobacterium* strains found in primary tumors and paired metastases, suggesting that the tumor microbiota may be an important component of the tumor microenvironment and involved in tumor metastasis [66].

As for the specific mechanism of microbiome dysbiosis in lung tumorigenesis, there are some studies that give us some supporting data. For example, oral taxa were found to be most abundant in the lower respiratory tract of patients with lung cancer, including *Streptococcus* and *Veillonella* [57,67,68]. In vitro, it was noted that the exposure of airway epithelial cells to *Streptococcus*, *Prevotella*, and *Bacteroides* can upregulate ERK (extracellular signal-regulated kinase) and phosphatidylinositol 3-kinase (PI3K) signaling pathways. The γδ T cells are the major tissue-resident T-cell component of mucosal barrier tissues. They have ‘‘innate-like’’ characteristics and respond rapidly to infections and tissue damage [61]. Another study found that the lung microbiome contributes to the inflammation associated with lung adenocarcinoma through activating γδ T cells within the lungs. Specifically, lung bacteria stimulated myeloid cells to produce MyD88-dependent IL-1b and IL-23, which induced the proliferation and activation of γδ T cells and production of effector molecules such as IL-17, thereby promoting inflammation and tumor cell proliferation [61,69].

## 5. The Lung Microbiome and Multiple Sclerosis

Numerous studies have demonstrated that the bidirectional interaction between the gut microbiome (GM) and the central nervous system—the so-called gut microbiome–brain axis (GMBA)—deeply affects brain function and is one of the critical regulators of the brain’s immune response [70]. Just as the communication between the gut and brain is called the “gut–brain axis”, the biological pathway of communication between the lungs and the brain can also be called the lung–brain axis. In fact, there is an inherent connection between the respiratory system and the central nervous system in both healthy and diseased states. For instance, it has been shown that pulmonary complications such as neurogenic pulmonary edema (NPE), acute respiratory distress syndrome (ARDS), and ventilator-associated pneumonia (VAP) are likely to occur after traumatic brain injury (TBI), and brain hypoxia and intracranial hypertension are also likely to occur after lung injury [71,72,73,74]. Specifically, the lungs and brain communicate with one another through triggered inflammatory factors or specific signaling pathways in response to traumatic stimuli, causing corresponding pathological changes. Recent research suggests that lung microbes may also be a bridge between the lungs and the brain. In a recent issue of *Nature*, Hosang et al. [13] demonstrated that the lung microbiome regulates the immune reactivity of the central nervous tissue, thereby influencing its susceptibility to autoimmune disease development. This provides a potential treatment for autoimmune diseases of the brain, such as multiple sclerosis.

Multiple sclerosis (MS) is a central nervous system autoimmune disease that is most likely caused by a combination of genetic and environmental factors [14,75,76]. The only dominant risk gene found to date is the major histocompatibility *gene* variant HLADR2/15—a *gene* complex that strongly influences T-cell immune reactivity [75,77]. Environmental factors include smoking, sunlight, EB virus infection, and obesity [75,78]. Many clinical studies have shown that the immune system—particularly autoimmune T cells such as Th1, Th17, CD4+ T, and CD8+ T cells—plays an important role in multiple sclerosis. Multiple sclerosis is thought to be caused by the activation of peripheral autoreactive CD4+ T cells. Activated CD4+ T cells can differentiate into Th1 and Th17 cells that migrate into the central nervous system and initiate the disease process. In addition, CD4+ T cells can enhance the activity of CD8+ T cells and other immune cells to protect against central nervous system myelin and other structures [75,76,77,79,80].

Although the healthy lung possesses low microbial biomass, dysbiosis of the lung microbiome is also associated with local inflammation and tissue damage—particularly in chronic lung diseases [14]. However, it remains unclear whether the lung microbiome can have an impact on distal tissue immunity and homeostasis. Before entering the central nervous system, autoreactive T cells temporarily reside in the lung tissue, flowing along the airway toward the bronchi-associated lymphoid tissue and draining the mediastinal lymph nodes. Eventually, they enter the central nervous system through the blood circulation [14,81]. This suggests that our lungs may be involved in the central nervous system’s autoimmune processes. Recently, there has been a groundbreaking study about the lung microbiome and multiple sclerosis, investigating whether disruption of the lung microbiome with the antibiotic neomycin affected the severity of experimental autoimmune encephalomyelitis (EAE) in rats (an animal model of multiple sclerosis) [13].

First, in the rat EAE model, the authors found that intratracheal administration of neomycin decreased the diversity and increased the abundance of the lung microbiome by 2.5-fold compared to the control group. However, the gut microbiota did not change. In addition, the incidence of the EAE model was significantly reduced in neomycin-treated rats with intravenous myelin basic protein (MBP)-specific T cells (TMBP) [13]. The study also demonstrated that neomycin did not affect the proliferation of effector T cells and their encephalitis-causing potential. In the EAE model, the expansion of effector T cells and their ability to enter the blood were not affected by the labeling of effector T cells. Moreover, the interaction between effector T cells and endothelial cells of the blood–brain barrier (BBB) was not affected [13]. In bacterial transfer experiments, clinical symptoms of EAE were found to be significantly reduced in rats receiving a BALF microbiome from neomycin-treated rats. Taken together, these results suggest that neomycin protects against EAE by altering the lung microbiome, rather than through the proliferation and migration of T cells [13]. One of the most important findings was that the dysbiosis of the lung microbiome was linked to changes in the microglia [13]. Microglia are the primary resident immune cells in the central nervous system and are sensitive to T-cell-derived stimuli. Studies have shown that microglial dysfunction is associated with the occurrence and progression of several neurodevelopmental and neurodegenerative disorders. Notably, a shift in reactive microglia to type I IFN immune responses may modulate microglial responsiveness to type II IFN-dominated autoimmune challenges [13]. This study still found that the microglia in rats pretreated with neomycin showed reduced activation and a shift toward type I interferon (IFN) signaling after local stimulation with pro-inflammatory cytokines. To further explore the relationship between the microglia and the lung microbiome, the author focused on the changes in the lung microbiome after treatment with neomycin. Gram-negative *Bacteroidetes* were discovered to be the most abundant phylum of bacterial phyla (37%), with a 2.5-fold increase in number in the neomycin-treated group compared to the phosphate-buffered saline (PBS)-treated group. It is well known that LPS is one of the components of the bacterial cell wall that can induce a type I IFN response. The levels of LPS in bronchoalveolar lavage fluid were significantly increased in neomycin-treated rats. Polymyxin B, a polypeptide antibiotic that can neutralize LPS, increased the severity of EAE. However, direct injection of LPS into the lungs or brain significantly ameliorated EAE [13].

The study concluded that neomycin treatment shifted the composition of the lung microbiome to a higher abundance of bacteria that produce more LPS, which crossed the blood–brain barrier into the central nervous system to promote the production of type I IFN in the microglia and, thus, played a neuroprotective role [13,14,82,83]. The study initially revealed a mechanism for the involvement of the lung microbiome in the process of brain autoimmunity. This novel concept—the lung–brain axis—requires more research to define and validate it.

## 6. Conclusions and Perspectives

Owing to the development of culture-independent techniques, a profound understanding of the microbiome in the lungs has become possible. The diversity and abundance of the lung microbiome are less than those of the gut microbiome, but the lung microbiome is altered in many diseases. Studies have shown that the lung microbiome is closely associated with many chronic lung diseases, including chronic obstructive pulmonary disease, asthma, and lung cancer [3,41,84,85]. In these diseases, the microbial load and diversity in the lungs are altered, and the abundance of some specific bacteria either increases or decreases [41,54,68]. It is suggested that the lung microbiome plays an important role in contributing to the occurrence and development of lung diseases. As the research advances, it has been found that an intimate relationship exists between the lung microbiome and multiple sclerosis. Studies have shown that LPS can cross the blood–brain barrier (BBB) into the brain through the blood circulation, further influencing the development of multiple sclerosis by regulating the microglia in the brain [14,82,83]. This suggests that, as with the gut–brain axis, there is a close connection between the lungs and the brain, which we can also call the lung–brain axis. It is clear that the gut–brain axis is a complete biochemical signaling communication network through neuromediators involving the vagus nerve and the hypothalamic–pituitary–adrenal axis [78,86]. Nevertheless, there are still many unanswered questions on the lung–brain axis. A study by Hosang et al. [13] posited a reasonable mechanism for the regulation of brain autoimmunity by the lung microbiome, opening up new avenues to explore possible new therapeutic interventions. In addition, the role of the lung microbiome in other brain diseases—such as Parkinson’s disease, Alzheimer’s disease, intracerebral hemorrhage, and glioma—has yet to be studied.

Previous studies of the lung microbiome were mainly correlational and descriptive studies, which could only illustrate the relationship between the lung microbiota and disease, but could not reveal whether there was a causal relationship between the two. More longitudinal human intervention and mechanical animal studies are needed to address some key issues and causality. The following questions remain to be answered: (1) Is the microbiome a cause or a consequence (a marker) of disease, or both? (2) What are the specific mechanisms by which the lung microbiome affects the host? (3) In addition to the lung microbiome, do derivatives of lung microbes influence the host? (4) Could manipulating the lung microbiome represent the future of disease treatment? As we know, the gut microbiota can affect the brain function through microbiota-derived metabolites, immune and inflammatory pathways, and the microbiota themselves. Based on these factors, it is hypothesized that the lung microbiota may affect brain autoimmune diseases through the abovementioned mechanisms, as shown in Figure 2. Studies on the treatment of disease by interfering gut microbes have yielded fruitful achievements, including probiotics, prebiotics, antibiotics, and fecal bacterial transplantation. It is believed that, in the near future, modifying microbe–host interaction could be a therapeutic strategy to treat lung diseases and brain autoimmunity.

## Figures and Tables

**Figure 1 ijms-24-02170-f001:**
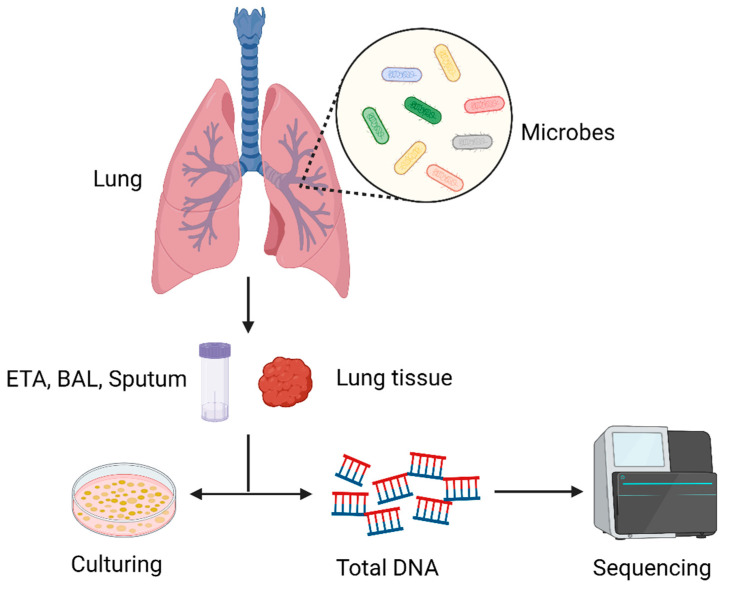
Schematic diagram of the process of lung microbiome detection: First, a qualified sample needs to be collected. Common samples include endotracheal aspirates (ETAs), bronchoalveolar lavage (BAL), and sputum, but in the case of diagnosed lung cancer, lung biopsy tissue can also be used as the sample to be detected. Next, there are two main types of analysis methods: culture-based and culture-independent methods; the former refers to the isolation of microbes by selecting suitable media and culturing them in a suitable environment, while the latter refers to the extraction of DNA from microbes for sequencing to obtain microbial profiles.

**Figure 2 ijms-24-02170-f002:**
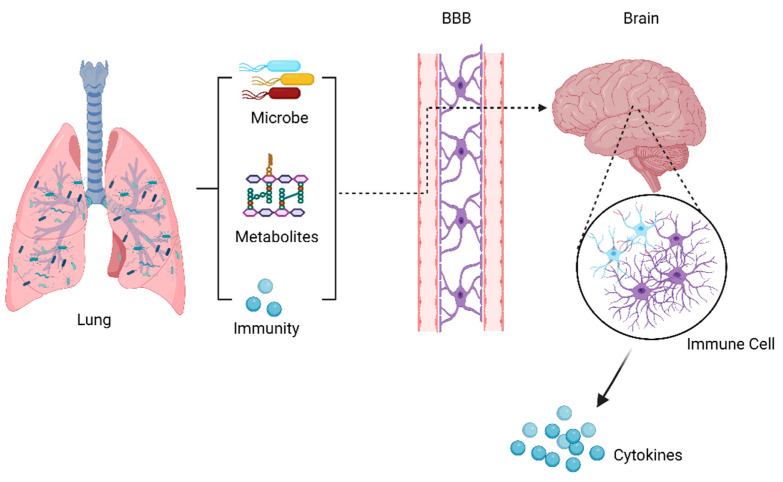
Possible communication pathways in the lung–brain axis. This communication between the lungs and the brain takes place through microbiota, metabolic, and immunological pathways, which may cross the BBB and reach the CNS to alter immune cells in the brain, influencing the brain disease.

## Data Availability

Not applicable.

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
