# Peer review of "The Lung Microbiome: A New Frontier for Lung and Brain Disease"

_ijms, 2023, doi:10.3390/ijms24032170_

Round 1

Reviewer 1 Report

Abstract

Line 34

What do the authors mean by low-level microbiomes?

Line 36

Explain what do you mean by local replication.

Line 94 

delete "over many years"

Lines 95-96 

Rewrite the sentence "Metatranscription........."

Lines 101-104 - Rewrite these lines

Line 146, add word "the" before disease

Line 173 ref number not added after Yan et al. [xx]

Line 193 No proper in text reference 

Line 196 Ref number should be put after et al.[xx]

Line 206 T. Whipplei should be italicized 

Line 226 Ref number should be put after et al.[xx]

Line 228 correct in-text reference 

Line 223 Peters Reference number missing 

Line 237 Reference format not correct 

Lines 241-242 Rephrase these lines; these are poorly written.

Line 278 Correct the reference Hosang

Headings are repetitive- 5, 5.1 and 5.2 almost discuss the same thing.

Line 305 correct the reference 

Line 332-334- Rephrase- not well  written 

Apart from these

There are a lot of repetitive sections throughout the manuscript. 

The authors should stick to the title and include relevant sections only.

The Language of the manuscript should be corrected.  

Reviewer 2 Report

Some minor corrections are needed, especialy refferences.

- It is nor an original research but it addresses an important question regarding lung microbiome and diseases like lung cancer and multiple sclerosis

  -The topic is original and the lung-brain axis is a new concept that is recently studied

-The manuscript is a review of relevant papers in field

-The method of literature review should be mentioned if it is the case.

-Please include any additional comments on the tables and figures.

Reviewer 3 Report

Review report - 

Information is limited about the role of lung microbiome in Lung and Brain Disease. Thanks to the authors for choosing the topic "The Lung Microbiome: A New Frontier for Lung and Brain Disease" as review. Overall, I liked the article, but in-depth discussion about the available results and all possible emerging hypotheses (about the role of lung microbiome in disease development) would have better served the purpose. I have a few comments as given below.    In line 116, what do you mean by "Due to immortality of lung biopsy"?   Line 138 - "However, studies have shown less variability in the lung microbiome" Comment - Please put appropriate citation/s.   In line 203, What do you mean by "neutrophil patients" (all of a sudden)?    Line 227 - Please abbreviate SPF?   In line 159-160 - " ...the predominant bacterial genera were Moraxella catarrhalisStreptococcus pneumoniae, and Pseudomonas aeruginosa" Comment - These are bacterial species not bacterial genera.   In line 165-167 "acute exacerbations of COPD were associated with a significant increase in Proteobacteria and a significant reduction in Bacteroidetes and Firmicutes"  Comment - How to interpret the result mentioned above to a fruitful hypothesis for future studies? Please discuss.    line 175-177 "It has been shown that relative abundances of Moraxella catarrhalis and Pseudomonas aeruginosa were most increased in COPD, while Prevotella intermedia was most depleted in COPD compared to healthy individuals".  Comment - More discussion is required regarding the role of above-mentioned bacterial species in COPD development.    Line 193-196 - "In addition, compared with controls, the bacterial burden of the lung microbiome was elevated in patients with asthma, with a marked increase in the number of Proteobacteria containing important potential pathogens Haemophilus, Moraxella, and Neisseria [54]. In addition to bacterial load, there is an increase in bacterial diversity in asthma patients". Comment - Please discuss if these above-mentioned pathogens (Haemophilus, Moraxella and Neisseria) play a role or what kind of role they might play in asthma development?    In line 260-263 "Specifically, lung bacteria stimulated myeloid cells to produce MyD88-dependent IL-1b and IL-23, which induced proliferation and activation of V6+V1+ T cells and production of the effector molecule, such as IL-17, thereby promoting inflammation and tumor cell proliferation". Comment - Please discuss a bit about MyD88, IL-1b, IL-23, V6, V1 so that the above sentences could make more sense.   Line 343-344 "One of the most important findings was that the dysbiosis of the lung microbiome was linked to changes in microglia". Comment - Please put citation/s.   Line 359-360, "However, direct injection of LPS into the lung or brain significantly ameliorated EAE" Comment- Please put citation/s.

Round 2

Reviewer 1 Report

Response 2: Thank you very much for your suggestions. Please forgive us for not expressing ourselves clearly. We have rewritten the sentence.

Original sentence: Moreover, immune factors and local replication are also associated with the lung microbiome.

Revised sentence: In addition, the composition of the lung microbiome is influenced by the relative reproduction rate of the microbiome and immune factors.

Correct: Elimination, immigration, and relative growth within its communities all contribute to the composition of the lung microbiome.

Original sentence: Traditional culture-based methods for assessing bacterial viability have been proven to be effective and cost-effective over many years.

Revised sentence: Traditional culture-based methods for assessing bacterial viability have been proven to be effective and cost-effective.

Correct: Traditional culture-based methods for assessing bacterial viability have been proven to be cost-effective.

Original sentence: Metatranscriptomics may therefore be more efficient because it is dependent on living cells.

Revised sentence: This is where metatranscriptomics comes into its own, as it studies the gene expression of microbes within natural environments.

Correct: Metatranscriptomics is more suitable here, as it studies the gene expression of microbes within natural environments.

Point 5: Lines 101-104  Rewrite these lines

Response 5: Thank you for your valuable suggestions. We have rewritten the sentence.

Original sentence: Recent years have seen an increase in studies on the lung microbiome, but both the number and quality of that research have lagged behind that on the gut microbiota. Traditionally, the normal lung, including the alveoli below the vocal cords to the distal alveoli, is considered to be in a sterile environment.

Revised sentence: Numerous studies have been conducted on the effects of gut bacteria on human disease and health, and in recent years, the lung microbiome has attracted increased interest. Traditionally, it was believed that a healthy lung, including the alveoli below the vocal cords and the distal alveoli, was a sterile organ.

Correct: Numerous studies have been conducted on the effects of gut bacteria on human disease and health, and in recent years, the lung microbiome has attracted increased interest. Traditionally, it was believed that a healthy lung, including the distal alveoli, was a sterile organ.

Original sentence: Many studies have shown that the lung microbiome can result in disease or increase disease susceptibility.

Revised sentence: Many studies have shown that the lung microbiome can result in disease or increase the disease susceptibility.

Correct: Many studies have shown that the lung microbiome can result in disease or increase disease susceptibility.

These headings in the revised draft are as follows:

5. Lung Microbiome and Multiple Sclerosis

5.1 The lung-brain axis.

5.2 The Lung Microbiome in multiple Sclerosis

 Again the headings 

5 and 5.2 are same ??

check all the references carefully, recheck
